# A Comparative Study of New Fluorescent Anthraquinone and Benzanthrone α-Aminophosphonates: Synthesis, Spectroscopy, Toxicology, X-ray Crystallography, and Microscopy of *Opisthorchis felineus*

**DOI:** 10.3390/molecules29051143

**Published:** 2024-03-04

**Authors:** Armands Maļeckis, Marija Cvetinska, Muza Kirjušina, Ligita Mežaraupe, Sanita Kecko, Inese Gavarāne, Vladimir Kiyan, Lyudmila Lider, Veronika Pavlova, Marina Savicka, Sergey Belyakov, Elena Kirilova

**Affiliations:** 1Institute of Chemistry and Chemical Technology, Faculty of Natural Sciences and Technology, Riga Technical University, P. Valdena Str. 3, LV-1048 Riga, Latvia; 2Department of Ecology, Institute of Life Sciences and Technology, Daugavpils University, LV-5401 Daugavpils, Latvia; 3Laboratory of Biodiversity and Genetic Resources, National Center for Biotechnology, 13/5 Kurgalzhynskoye Road, Astana 010000, Kazakhstan; 4Faculty of Veterinary Medicine and Animal Husbandry Technology, S. Seifullin Kazakh Agro Technical Research University, 62 Zhenis Avenue, Astana 010011, Kazakhstan; 5Latvian Institute of Organic Synthesis, Aizkraukles Str. 21, LV-1006 Riga, Latvia; 6Department of Environment and Technologies, Faculty of Natural Sciences and Healthcare, Daugavpils University, LV-5401 Daugavpils, Latvia

**Keywords:** fluorescence, solvatochromism, aminophosphonate, benzanthrone, anthraquinone, Kabachnik–Fields reaction, one-pot reaction, crystal structure, toxicology, confocal laser scanning microscopy

## Abstract

In this research, we explore the synthesis of and characterize α-aminophosphonates derived from anthraquinone and benzanthrone, focusing on their fluorescence properties and potential applications in confocal laser scanning microscopy (CLSM). The synthesized compounds exhibit notable solvatochromic behavior, emitting fluorescence from green to red across various solvents. Spectroscopic analysis, including ^1^H-, ^13^C-, and ^31^P-NMR, FTIR, and mass spectrometry, confirms the chemical structures. The compounds’ toxicity is evaluated using etiolated wheat sprouts, revealing varying degrees of impact on growth and oxidative damage. Furthermore, the study introduces these α-aminophosphonates for CLSM imaging of the parasitic flatworm *Opisthorchis felineus*, demonstrating their potential in visualizing biological specimens. Additionally, an X-ray crystallographic study of an anthraquinone α-aminophosphonate provides valuable structural insights.

## 1. Introduction

Fluorescence is a phenomenon wherein a substance absorbs light and promptly releases it at a longer wavelength, occurring within nanoseconds and resulting in a spectral shift towards longer wavelengths. An effective organic fluorophore possesses specific characteristics. A crucial factor is a high absorption coefficient, ensuring efficient molecule excitation and, consequently, brighter emitted fluorescence. The Stokes shift, representing the difference between excitation and emission wavelengths, should be sufficiently large to minimize reabsorption and maximize sensitivity. Additionally, an efficient fluorophore exhibits a high quantum yield, signifying that most absorbed energy is converted into emitted light rather than being redirected into competing non-radiative processes. Photostability is equally vital in order to prevent degradation or loss of fluorescent properties over time or with repeated exposure to light. These key attributes collectively define an optimal organic compound for fluorescence applications [1].

In our prior investigation of α-aminophosphonates, we explored derivatives belonging to the anthrone dye family, specifically focusing on anthraquinone and benzanthrone derivatives [2,3]. These compounds, characterized by a donor–π–acceptor architecture, have attracted considerable attention owing to the noteworthy properties mentioned earlier.

While both anthraquinone and benzanthrone derivatives have held a crucial position in advancing the dye industry and are typically employed in vat dyeing techniques [4,5], various other applications have arisen. For anthraquinones, these include: serving as emitters in organic light-emitting diodes (OLEDs) [6,7], detecting ionic species in chemosensory processes [8,9], contributing to cellular imaging [10,11], and finding utility in the medical field as agents with fungicidal, antibacterial, insecticidal, antiparasitic, antiviral, and anticancer properties [12]. On the other hand, benzanthrones are suited for diverse uses: in liquid crystal displays [13,14], polymeric materials [15], as probes for the pH of solutions and cations [16,17], selective detectors of amyloid fibrils of the lysozyme enzyme [18], and visualization of parasitic trematodes and nematodes through confocal laser scanning microscopy imaging [19].

α-aminophosphonates, analogues of amino acids, possess the ability to inhibit various enzymes involved in amino acid metabolism, thus acting as antagonists and displaying a diverse range of biological activities such as antifungal, antimicrobial, antiviral, anticancer, herbicidal, and neuromodulatory effects [20,21]. Additionally, these compounds, both natural and synthetic, have been extensively studied for their potential applications as lubricating additives [22], antioxidants [23], sorbents [24,25], and corrosion inhibitors [26,27]. Furthermore, α-aminophosphonates derived from aromatic amines, including benzene, naphthalene, anthracene, pyrene, and phenanthrene, as well as bis-aminophosphonates with anthracene rings, have been explored for their fluorescent properties [28,29,30,31,32,33].

In our previous research, we have independently explored some benzanthrone and anthraquinone α-aminophosphonates. The difference between these compounds has not been explored yet. In this study, we synthesize novel derivatives by utilizing aldehydes featuring previously unreported substituents and bridge the gap by comparing their synthesis processes; luminescent properties; toxicological aspects, which have not been previously analyzed for anthraquinone derivatives; and applicability for confocal laser scanning microscopy, an aspect overlooked in studies of benzanthrone derivatives. An X-ray crystallographic study of an anthraquinone α-aminophosphonate, absent in earlier research, and benzanthrone α-aminophosphonate was performed as well.

## 2. Results and Discussion

### 2.1. Synthesis

The synthesis of target compounds was carried out using the Kabachnik–Fields reaction conditions. Numerous reviews have elucidated the mechanism and summarized diverse approaches for this three-component reaction [34,35,36,37]. These approaches encompass catalyzed and catalyst-free variations, as well as synthesis involving the use of ionic liquids, dehydrating agents, and microwave irradiation. We utilized our previously outlined synthetic procedure wherein an aromatic amine reacts at an elevated temperature with suitable aromatic aldehyde in excess, employing a dimethylphosphonate as both the reactant and the solvent. This one-pot, solvent-free approach is also influenced by the convenience of the subsequent extraction and purification of α-aminophosphonates. An excess of an aldehyde and a dimethylphosphonate undergo hydrolysis under moderately basic conditions, leading to the isolation of a pure compound through simple recrystallization. Compounds were obtained, on average, in 65% yields. Although the substrate has no significant impact on the reaction yield, the reaction with 4-bromobenzaldehyde, on average, yielded more than 70%. In contrast, the reaction with 4-(methylthio)benzaldehyde resulted in a slightly lower yield of 66%, and the lowest yield of 57% was achieved with 3-cyano-4-fluorophenylbenzaldehyde. Compounds were synthesized according to Figure 1.

### 2.2. Spectroscopy

#### 2.2.1. Structure

To validate the chemical structures of the acquired compounds, ^1^H-, ^13^C-, ^31^P-NMR, and FTIR spectroscopy, as well as mass spectrometry, were performed.

The obtained infrared spectra showed broad amino group (NH) stretching vibration bands around 3225–3237 cm^−1^ for anthraquinone derivatives and around 3297–3484 cm^−1^ for benzanthrone derivatives; two stretching vibration peaks of anthraquinone carbonyl groups (C=O) and one for benzanthrone carbonyl group at around 1630–1669 cm^−1^; and broad aliphatic and aromatic carbon–hydrogen (C–H) stretching vibration bands around 2848–3077 cm^−1^.

The confirmation of the obtained compounds’ structures was achieved through ^1^H-NMR spectroscopy. The characteristic signals of aromatic protons were identified. The α-carbon hydrogen signal manifested as a doublet of doublets, arising from coupling with phosphorus. Hydrogens within the methyl groups of the phosphonate moiety appeared as distinct multiplets due to their magnetic non-equivalence. In APT NMR spectra, identifiable peaks of carbonyl group carbons emerged at approximately 184 ppm. Furthermore, the APT spectra exhibited distinct doublets due to carbon–phosphorus coupling and the magnetic non-equivalence of carbon atoms within the phosphonate moiety. In ^31^P-NMR spectra, phosphorus peaks appeared as singlets at around 23 ppm (decoupled mode) and as doublets for fluorine-containing compounds **2c** and **4c**. The acquired data align well with previously reported NMR results for analogous compounds: benzanthrones [38,39], anthraquinones [40], and α-aminophosphonates [41,42,43].

The mass of the synthesized compounds, determined through the results obtained from high-resolution mass spectrometry, was verified to match the calculated values.

#### 2.2.2. Photophysical Properties

To investigate the photophysical characteristics of the synthesized compounds, UV–Vis absorption and fluorescence emission spectra in solvents of varying polarity (benzene, chloroform, ethyl acetate (EtOAc), acetone, DMF (*N*,*N*-dimethylformamide), DMSO (dimethyl sulfoxide), and ethanol (EtOH)) were acquired.

The examined dyes demonstrated fluorescence and exhibited significant solvatochromic behavior, emitting light ranging from green in hexane to red in ethanol. The solvatochromic properties of these derivatives indicate that the fundamental mechanism governing fluorescence was internal charge transfer during excitation, specifically from the electron-donating amino group to the electron-withdrawing carbonyl groups. Table 1, Table 2 and Table 3 provide a summary of data representing absorption maxima, molar attenuation coefficients, fluorescence maxima, fluorescence quantum yields, and Stokes shifts.

In solutions, anthraquinone α-aminophosphonates **2a**, **2b**, and **2c** exhibit broad band absorbance around 465–488 nm and fluorescence from 584 nm (benzene) to 628 nm (ethanol) (Figure 2), whereas benzanthrone α-aminophosphonates **4a**, **4b**, and **4c** absorb around 447–498 nm and fluorescence from 516 nm (hexane) to 636 nm (ethanol) (Figure 3).

While extinction coefficients, on average, were slightly higher for benzanthrone compounds than anthraquinone ones with the same substituents on the phenyl ring, fluorescence quantum yields were considerably higher, indicating a more efficient conversion of absorbed photons into emitted fluorescence. Because of the diminished fluorescence intensity of the examined anthraquinone dyes, the emission spectra exhibited noise. The polarity effect of the medium on fluorescence was more pronounced than on the absorption for both substrates. Among the studied dyes, the highest Stokes shift of 4997 cm^–1^ was observed for the anthraquinone derivative **2c** in ethanol. The introduction of strong electron-accepting groups (**2c** and **4c**) significantly decreased both the extinction coefficients and quantum yields. The absorption spectra of the recently examined benzanthrone alkyl and aryl sulfides spanned from 403 to 448 nm [44], and amidines absorbed in the range of 410 to 495 nm [45,46], whereas 3-substituted benzanthrone amino derivatives exhibited absorption between 430 and 520 nm [47]. For 3-amino-9-nitrobenzanthrone derivatives, absorption occurred in the range of 450 to 560 nm [48]. This suggests that the benzanthrone amino group, when attached to a phosphoryl group, demonstrates a stronger donating effect compared to amidino and thiol groups, albeit to a slightly lesser degree than an alkyl amino group.

The molecular–microscopic solvent polarity parameter ($E_{T}^{N}$) considers the molecular aspects of the solvent structure that contribute to its overall polarity. Plots (Figure 4) show how changes in solvent polarity affected the emission wavelength maxima of compounds **2a** (anthraquinone derivative) and **4a** (benzanthrone derivative). As the polarity of the solvent rose, there was a corresponding lengthening of the emission wavelength. The linear correlation observed between the $E_{T}^{N}$ parameter and emission maxima suggests the presence of general solute–solvent interactions, including H-bonding interactions, in the majority of solvents. The influence of polarity on the emission wavelength maxima was more pronounced for compound **4a** compared to **2a**.

### 2.3. Toxicity Assessment

For the evaluation of the toxicity of compounds **2a** and **4a**, 6 × 10^–5^ M and 3 × 10^–6^ M solutions were prepared. Within 24 h, the etiolated wheat sprouts of the cultivar “Fenomen” were grown in a dark chamber at a temperature of 26 °C on moist filter paper. Wheat is a convenient object for research in plant physiology and biochemistry because the development of cereals is synchronous throughout ontogeny [49]. The first leaf and coleoptile were used because the first leaf is a developing organ of wheat, but the coleoptile is a senescent organ of wheat, and the processes which occur in these organs are different. A morphology test on filter paper was carried out in plastic dishes with three layers of paper on the bottom. Each dish contained 5 mL of dye solution or 5 mL of distilled water (control) and 30 seeds, which were covered by a lid. Dishes containing seeds were incubated for 5 days in a dark chamber at 26 ± 1 °C. The lengths of the first leaf and coleoptile were measured in millimeters (mm). Data are presented taking into consideration the standard error (SE) of three replicates.

The highest concentration of both dyes had a significant toxicological effect on the growth of the wheat leaves and coleoptiles (Figure 5); thus, an evaluation of the toxicity of compounds **2a** and **4a** on the wheat seedlings was performed using only 3 × 10^–6^ M dye concentration. As a result, the phytotoxicity of the **2a** on the first leaf and coleoptile was higher than that of the **4a** (Table 4). We used this method to assess oxidative damage in wheat grown at lower concentrations as well (Table 5). The MDA levels were raised in the wheat leaves and coleoptile with both dyes compared to the control. This increase suggests that the cell membranes in wheat leaves and coleoptiles were experiencing oxidative damage, leading to lipid peroxidation. Nevertheless, the electrolyte leakage was decreased in the wheat sprouts. The decrease in electrolyte leakage suggests that the cell membrane’s integrity was maintained or repaired to some extent, possibly due to the plant’s defense mechanisms. The increase in MDA levels and the decrease in electrolyte leakage are indicative of complex cellular responses to oxidative stress due to dye toxicity.

Both dyes were found to cause significant oxidative damage to wheat seedlings, as evidenced by elevated levels of malondialdehyde (as the indicator of oxidative stress) content and electrolyte leakage compared to control wheat sprouts.

The results thus show that the toxic properties of the studied substances differed: the anthraquinone derivative **2a** was more toxic to plants than analogous benzanthrone one **4a**. This may arise from the quinone nature of **2a**, which, when reacting with antioxidants, leads to the release of reactive oxygen species (ROS) [50].

Similar data were obtained for benzanthrone derivatives with the thiophene and unsubstituted phenyl group [2].

### 2.4. Confocal Laser Scanning Microscopy Imaging of Trematode Opisthorchis Felineus

*Opisthorchis felineus* is a parasitic flatworm, commonly known as the cat liver fluke, that primarily infects the livers and bile ducts of various mammals, including humans. The life cycle of *O. felineus* involves several stages and includes both intermediate and definitive hosts. Adult flukes residing in the bile ducts release eggs, which are then excreted with the host’s feces. Embryonated eggs release miracidia upon reaching the external environment. Miracidia penetrate specific freshwater snails (e.g., *Radix* spp.) for further development [51]. Within the snail host, miracidia transform into sporocysts and then rediae, undergoing asexual reproduction. Cercariae, free-swimming larvae, are released from the snails. Cercariae infect freshwater fish, the second intermediate hosts [52]. Cercariae encyst in the muscles of fish, becoming metacercariae, the infective stage for the definitive host. Definitive hosts, including mammals and humans, become infected by consuming raw or undercooked fish containing metacercariae [53]. The life cycle is completed when adult flukes release eggs into the bile ducts, restarting the cycle. *Opisthorchis felineus* is endemic in regions around the Baltic Sea, Caspian Sea, and Black Sea [54]. Various estimates suggest that the current infection rate with these liver flukes potentially affects up to 40 million people, while a substantial risk group of approximately 600–750 million people exists in Eurasian countries [55]. These parasites are expanding their colonization into previously unreported regions in Eurasia [56]. Opisthorchiasis in humans is associated with chronic liver disease and an increased risk of cholangiocarcinoma [57]. Moreover, *O. felineus* can also infect domestic and wild mammals, which serve as reservoir hosts and contribute to the persistence of the parasite in certain ecosystems [58]. To bring all of this together, *O. felineus* poses a significant public health concern. Thus, understanding the life cycle and epidemiology of *O. felineus* is crucial for implementing effective control strategies and public health interventions in endemic areas.

Among the synthesized anthrone-based dyes investigated in this study, four specific compounds were subjected to testing with *Opisthorchis felineus*. Two of these compounds, namely, **4b** and **4c**, were benzanthrone dyes, which have pronounced lipophilic properties, while the remaining two, **2b** and **2c**, were anthraquinone dyes, exhibiting comparatively lower hydrophobic characteristics. According to the initial data, **4b** and **4c** dyes showed the structure and muscle of the parasite *O. felineus* slightly more clearly (see Figure 6 and Figure 7). Thus, it can be concluded that benzanthrone dyes are better for visualizing *Opisthorchis felineus* flukes. The obviously hydrophilic nature of the dye has a significant impact on the visualization of these biological objects. Overall, all tested dyes demonstrated promising initial outcomes, prompting us to undertake a more comprehensive examination of these dyes in subsequent studies.

### 2.5. X-ray Crystallographic Study

Figure 8 shows a perspective view of molecule **2a** with thermal ellipsoids, and the atom-numbering scheme follows in the text. In the molecular structure, the planar anthraquinone and phenyl systems form a dihedral angle of 84.3(8)°. The torsion angle of C1–N11–C12–C19 is equal to 73.2(7)°. The coordination polyhedron of the phosphorus atom is a tetrahedron in which the central atom (P13) deviates by 0.463(6) Å from the plane of the base (O14, O15, O17). The molecular structure is stabilized by a strong intramolecular hydrogen bond of NH···O type between the amino group and the quinone oxygen O26; the parameters of this bond are as follows: N11···O26 = 2.636(7) Å, H11···O26 = 1.93(6) Å, N11–H11···O26 = 146(4)°.

The crystal structure of **2a** is achiral (space group is *P*
$\overset{-}{1}$); thus, despite the asymmetric carbon atom (C12), the substance represents a true racemate. A characteristic feature of the crystal structure of **2a** is the fact that there is a strong π–π stacking interaction between anthraquinone systems; the shortest intermolecular atom–atomic contact (C9a···C10) is equal to 3.332(8) Å. By means of these interactions, molecular stacks are formed in the crystal structure along the crystallographic direction [0 1 0] (Figure 9). The amino group is involved in the strong intramolecular H-bond; therefore, it does not form intermolecular H-bonds. However, there are carbon atoms with as increased electronegativity; first of all, these are carbons of methyl groups bonded to oxygen atoms. This is why moderated intermolecular H-bonds of CH···O type are formed in the crystal structure. The parameters of these bonds are as follows: C16···O27 = 3.292(9) Å, H16B···O27 = 2.31 Å, C16–H16B···O27 = 178°; C18···O17 = 3.244(9) Å, H18A···O17 = 2.50 Å, C16–H16B···O27 = 132°. It is also worth highlighting a rather strong halogen bond between bromine atoms with a length of 3.449(5) Å. Through this bond, the molecules in the crystal structure combine to form centrosymmetric dimers.

Figure 10 illustrates a perspective view of molecule **4a**, with thermal ellipsoids and the atom-numbering scheme following in the text. A characteristic feature of the crystal structure is static disorder of the dimethyl phosphonate group: In the crystals, there are two molecular forms of **4a**, which differ in the conformation of the dimethyl phosphonate group. The occupancy g-factors of these forms are 0.65 and 0.35; Figure 10 shows the form with the largest g-factor.

The peculiarity of the molecular structure is a fairly strong intramolecular hydrogen bond of NH···O type between the amino group and dimethylphosphonate. The parameters of this bond are as follows: N18···O28 = 2.947(4) Å, H11···O26 = 2.44(3) Å, N18−H18···O28 = 121(3)°. By means of this bond, an additional five-membered cycle is formed in the structure of **4a** molecules (see Figure 11). The H-bond stabilizes the molecular structure; through it, the oxygen atom O28 is not disordered, unlike the other atoms of the dimethyl phosphonate group.

The torsion angle of C3−N18−C19−C20 is equal to −66.8(3)°. In the molecular structure, the planar benzanthrone and phenyl systems form a dihedral angle of 73.7(3)°.

The crystal structure of **4a** is less dense than **2a**. The packing index for **4a**, calculated using Kitaigorodsky’s approach [59], is equal to 0.693, while for **2a**, it is 0.708. In the crystal structure **4a**, there is no halogen bond; only a weak intermolecular hydrogen bond of CH···O type with a length of 3.263(5) Å occurs between the C24−H24 group and oxygen atom O28. Numerous weak intermolecular CH···π interactions are also present, the strongest of which is C4−H4···*C_s_* (H4···*C_s_* = 3.14 Å), where *C_s_* represents the centroid of the phenyl ring. Figure 12 shows this interaction along with hydrogen bonds. In addition to **2a**, the crystal structure of **4a** is achiral (space group is *P*2_1_/*c*); thus, despite the asymmetric carbon atom (C19), the substance represents a true racemate.

## 3. Materials and Methods

### 3.1. Materials and Measurments

All of the reagents and solvents were obtained commercially and used without any additional purification. The assessment of the progress of reactions and purity of the synthesized compounds was performed using TLC on MERCK Silica gel 60 F254 plates, with hexane/acetone (1:2) as an eluent, and visualized under UV light. Melting points were determined using a METTLER TOLEDO™ Melting Point System MP70 apparatus. The IR spectrum was recorded on a Thermo Scientific Nicolet iS50 Spectrometer (ATR accessory; no. of scans: 64; resolution: 4; data spacing: 0.482 cm^−1^). ^1^H-, ^13^C-, and ^31^P-NMR were captured using a Bruker Avance 500 MHz (Bruker Corporation, Billerica, MA, USA) in CDCl_3_ at room temperature. Solvent peaks were used as the internal reference. Chemical shift (δ) values are reported in ppm. Accurate high-resolution mass measurements were conducted using the Orbitrap Exploris 120 (Thermo Fisher Scientific, Waltham, MA, USA), operating in Full Scan mode at a resolution of 120,000. The FLSP920 spectrofluorometer (Edinburgh Instruments Ltd., Livingston, UK) captured fluorescence emission spectra within the visible range of 450–800 nm, while the absorption spectra were acquired using the UV-visible spectrophotometer SPECORD^®^ 80 (Analytik Jena AG, Jena, Germany).

### 3.2. Synthesis and Characterization

General procedure for synthesis of derivatives **2a**–**2c** and **4a**–**4c**:In a 10 mL round-bottom flask fitted with a magnetic stirrer bar, 2 mmol of an amine, 6 mmol of an aldehyde, and 5 mL of dimethylphosphonate were combined. The mixture was stirred at 120 °C for 1 to 3 h, with progress monitored using TLC. After the completion of the reaction, the mixture was poured into 100 mL of concentrated sodium bicarbonate (NaHCO_3_) solution and left overnight while being stirred until a solid product was formed. The precipitate was then filtered, thoroughly washed with water, and dried. Purification of the product was achieved through multi-solvent recrystallization using xylenes and hexane.*Dimethyl ((4-bromophenyl)((9,10-dioxo-9,10-dihydroanthracen-1-yl)amino)methyl)phosphonate* (**2a**) was obtained as an orange compound in a 71% yield with a m.p. of 166 °C. R*f* = 0.72 (hexane-acetone, *v*/*v* 1:2). ^1^H NMR (500 MHz, *Chloroform-d*) δ 10.64 (dd, J = 10.4, 7.4 Hz, 1H, NH), 8.31 (dd, J = 7.7, 1.6 Hz, 1H_Ar_), 8.18 (dd, J = 7.6, 1.6 Hz, 1H_Ar_), 7.73 (td, J = 7.5, 1.5 Hz, 1H_Ar_), 7.68 (td, J = 7.5, 1.5 Hz, 1H_Ar_), 7.59 (d, J = 7.4 Hz, 1H_Ar_), 7.44 (d, J = 8.1 Hz, 2H_Ar_), 7.38 (t, J = 8.0 Hz, 1H_Ar_), 7.36–7.30 (m, 2H_Ar_), 6.75 (d, J = 8.6 Hz, 1H_Ar_), 4.92 (dd, J = 24.0, 7.3 Hz, 1H, NCH), 3.70 (ddd, J = 18.9, 10.7, 1.4 Hz, 6H, OCH_3_). ^13^C NMR (126 MHz, *Chloroform-d*) δ 185.86 (C=O), 183.41 (C=O), 149.92 (d, J = 13.5 Hz, C), 135.41 (CH), 134.82 (C), 134.67 (C), 134.12 (CH), 134.01 (d, J = 3.7 Hz, C), 133.44 (CH), 132.98 (C), 132.12 (d, J = 2.8 Hz, CH), 129.37 (d, J = 5.2 Hz, CH), 127.13 (CH), 126.85 (CH), 122.46 (d, J = 4.2 Hz, C), 118.24 (CH), 117.16 (CH), 114.82 (C), 54.60 (d, J = 152.5 Hz, NCH), 54.39 (d, J = 7.2 Hz, OCH_3_), 54.09 (d, J = 7.1 Hz, OCH_3_). ^31^P NMR (202 MHz, *Chloroform-d*) δ 22.38. FTIR (neat): 1631 and 1668 (C=O); 2852, 2905, 2952 and 3067 (CH); 3225 (NH). ESI-FTMS: calculated for [C_23_H_19_BrNO_5_P]: 500.0257, found: 500.0254.*Dimethyl (((9,10-dioxo-9,10-dihydroanthracen-1-yl)*amino*)(4-(methylthio)phenyl)methyl)phosphonate* (**2b**) was obtained as an orange compound in 63% yield with a m.p. of 154 °C. R*f* = 0.69 (hexane-acetone, *v*/*v* 1:2). ^1^H NMR (500 MHz, *Chloroform-d*) δ 10.63 (dd, J = 10.2, 7.4 Hz, 1H, NH), 8.31 (dd, J = 7.6, 1.6 Hz, 1H_Ar_), 8.18 (dd, J = 7.6, 1.6 Hz, 1H_Ar_), 7.72 (td, J = 7.5, 1.6 Hz, 1H_Ar_), 7.67 (td, J = 8.2, 6.7 Hz, 1H_Ar_), 7.58 (d, J = 7.4 Hz, 1H_Ar_), 7.41–7.33 (m, 3H_Ar_), 7.17 (d, J = 8.4 Hz, 2H_Ar_), 6.80 (d, J = 8.5 Hz, 1H_Ar_), 4.92 (dd, J = 23.6, 7.5 Hz, 1H, NCH), 3.71 (d, J = 10.6 Hz, 2H, OCH_3_), 3.66 (d, J = 10.7 Hz, 2H, OCH_3_), 2.39 (s, 3H, SCH_3_), 1.56 (d, J = 3.5 Hz, 2H, OCH_3_). ^13^C NMR (126 MHz, *Chloroform-d*) δ 185.75 (C=O), 183.49 (C=O), 150.16 (C), 150.06 (C), 138.99 (d, J = 3.8 Hz, C), 135.34 (CH), 134.75 (d, J = 3.6 Hz, C), 134.09 (CH), 133.36 (CH), 132.99 (C), 131.33 (d, J = 4.0 Hz, C), 128.14 (d, J = 5.3 Hz, CH), 127.12 (CH), 126.81 (CH), 126.67 (d, J = 2.7 Hz, CH), 118.44 (CH), 117.00 (CH), 114.70 (C), 54.65 (d, *J* = 153.0 Hz, NCH), 54.33 (d, J = 7.1 Hz, OCH_3_), 54.05 (d, J = 7.1 Hz, OCH_3_), 15.50 (SCH_3_). ^31^P NMR (202 MHz, *Chloroform-d*) δ 22.97. FTIR (neat): 1630 and 1669 (C=O); 2848, 2950, 2998 and 3077 (CH); 3237 (NH). ESI-FTMS: calculated for [C_24_H_22_SNO_5_P+H^+^]: 468.1029, found: 468.1017.*Dimethyl ((3-cyano-4-fluorophenyl)((9,10-dioxo-9,10-dihydroanthracen-1-yl)amino)methyl)phosphonate* (**2c**) was obtained as an orange solid in a 59% yield with a m.p. of 205 °C. R*f* = 0.71 (hexane-acetone, *v*/*v* 1:2). ^1^H NMR (500 MHz, *Chloroform-d*) δ 10.63 (dd, *J* = 10.7, 7.4 Hz, 1H, NH), 8.31 (d, *J* = 7.6 Hz, 1H_Ar_), 8.20 (d, *J* = 7.5 Hz, 1H_Ar_), 7.78–7.69 (m, 2H_Ar_), 7.69 (t, *J* = 4.0 Hz, 2H_Ar_), 7.64 (d, *J* = 7.4 Hz, 1H_Ar_), 7.42 (t, *J* = 8.0 Hz, 1H_Ar_), 7.16 (d, *J* = 8.4 Hz, 1H_Ar_), 6.68 (d, *J* = 8.5 Hz, 1H_Ar_), 4.95 (dd, *J* = 24.3, 7.3 Hz, 1H, NCH), 3.76 (dd, *J* = 12.9, 10.8 Hz, 6H, OCH_3_). ^13^C NMR (126 MHz, *Chloroform-d*) δ 186.14 (C=O), 183.23 (C=O), 163.94 (d, *J* = 3.3 Hz, C), 161.85 (C), 149.42 (d, J = 13.4 Hz, C), 135.59 (CH), 135.00 (C), 134.52 (C), 134.30 (dd, J = 8.5, 4.7 Hz, CH), 134.22 (CH), 133.66 (CH), 132.94 (C), 132.60 (C), 132.46 (d, J = 5.3 Hz, CH), 127.18 (CH), 126.93 (CH), 117.72 (CH), 117.58 (CH), 117.15 (dd, J = 20.0, 2.5 Hz, CH), 115.13 (C), 113.51 (C), 54.53 (d, J = 7.2 Hz, OCH_3_), 54.20 (d, J = 7.0 Hz, OCH_3_), 53.80 (d, J = 153.0 Hz, NCH). ^31^P NMR (202 MHz, *Chloroform-d*) δ 21.54 (d, J = 7.4 Hz). FTIR (neat): 1630 and 1667 (C=O); 2236 (C≡N); 2857, 2904, 2960 and 3046 (CH); 3235 (NH). ESI-FTMS: calculated for [C_24_H_22_SNO_5_P+H^+^]: 465.1010, found: 465.0996.*Dimethyl ((4-bromophenyl)((7-oxo-7H-benzo[de]anthracen-3-yl)amino)methyl)phosphonate* (**4a**) was obtained as a red solid in a 75% yield with a m.p. of 207 °C. R*f* = 0.68 (hexane-acetone, *v*/*v* 1:2). ^1^H NMR (500 MHz, *Chloroform-d*) δ 8.77 (dd, J = 7.3, 1.2 Hz, 1HAr), 8.37 (td, J = 7.9, 1.4 Hz, 2HAr), 8.09 (d, J = 8.3 Hz, 1H_Ar_), 8.04 (d, J = 8.2 Hz, 1H_Ar_), 7.75 (t, J = 7.8 Hz, 1H_Ar_), 7.61–7.54 (m, 1H_Ar_), 7.45 (d, J = 8.5 Hz, 2H_Ar_), 7.39–7.32 (m, 3H_Ar_), 6.42 (d, J = 8.3 Hz, 1H_Ar_), 5.87 (dd, J = 10.5, 6.7 Hz, 1H, NH), 4.93 (dd, J = 24.0, 6.6 Hz, 1H, NCH), 3.77 (d, J = 10.8 Hz, 3H, OCH_3_), 3.53 (d, J = 10.7 Hz, 3H, OCH_3_). ^13^C NMR (126 MHz, *Chloroform-d*) δ 184.01 (C=O), 143.65 (d, J = 14.3 Hz, C), 136.80 (C), 133.87 (d, J = 3.9 Hz, C), 133.25 (CH), 132.22 (d, J = 2.8 Hz, CH), 130.13 (CH), 129.67 (C), 129.23 (CH), 129.19 (CH), 129.16 (C), 128.75 (C), 127.98 (CH), 126.67 (CH), 126.05 (CH), 125.58 (CH), 123.40 (C), 122.53 (d, J = 4.2 Hz, C), 122.05 (CH), 117.48 (C), 106.92 (CH), 55.23 (d, *J* = 151.1 Hz, NCH), 54.28 (d, J = 7.0 Hz, OCH_3_), 54.08 (d, J = 7.0 Hz, OCH_3_). ^31^P NMR (202 MHz, *Chloroform-d*) δ 23.60. FTIR (neat): 1638 (C=O); 2849, 2952 and 3050 (CH); 3401 (NH). ESI-FTMS: calculated for [C_26_H_21_BrNO_4_P–H+]: 520.0318, found: 521.0320.*Dimethyl ((4-(methylthio)phenyl)((7-oxo-7H-benzo[de]anthracen-3-yl)amino)methyl)phosphonate* (**4b**) was obtained as a red solid in a 68% yield with a m.p. of 123 °C. Rf = 0.77 (hexane-acetone, *v*/*v* 1:2). ^1^H NMR (500 MHz, *Chloroform-d*) δ 8.77 (dd, J = 7.1, 0.8 Hz, 1H_Ar_), 8.41–8.35 (m, 2H_Ar_), 8.10 (d, J = 8.3 Hz, 1H_Ar_), 8.05 (d, J = 8.2 Hz, 1H_Ar_), 7.75 (t, J = 7.8 Hz, 1H_Ar_), 7.61–7.54 (m, 1H_Ar_), 7.42–7.32 (m, 3H_Ar_), 7.18 (d, J = 7.8 Hz, 2H_Ar_), 6.48 (d, J = 8.3 Hz, 1H_Ar_), 5.90 (s, 1H, NH), 4.94 (d, J = 23.8 Hz, 1H, NCH), 3.76 (d, J = 10.8 Hz, 3H, OCH_3_), 3.50 (d, J = 10.6 Hz, 3H, OCH_3_), 2.39 (s, 3H, SCH_3_). ^13^C NMR (126 MHz, *Chloroform-d*) δ 184.04 (C=O), 143.94 (d, J = 14.4 Hz, C), 139.20 (C), 136.89 (C), 133.22 (CH), 131.15 (d, J = 3.9 Hz, C), 130.10 (CH), 129.64 (C), 129.12 (C), 128.74 (C), 128.02 (CH), 127.97 (d, J = 1.6 Hz, CH), 126.78 (CH), 126.69 (d, J = 2.7 Hz, CH), 126.56 (CH), 126.19 (CH), 125.48 (CH), 123.41 (C), 122.02 (CH), 117.20 (C), 106.91 (CH), 55.24 (d, J = 151.8 Hz, NCH), 54.29 (d, J = 7.0 Hz, OCH_3_), 53.97 (d, J = 7.1 Hz, OCH_3_), 15.46 (SCH_3_). ^31^P NMR (202 MHz, *Chloroform-d*) δ 24.16. FTIR (neat): 1630 (C=O); 2851, 2914, 2951, 2982 and 3059 (CH); 3484 (NH). ESI-FTMS: calculated for [C_27_H_24_NO_4_PS+H+]: 490.1236, found: 490.1221.*Dimethyl ((3-cyano-4-fluorophenyl)((7-oxo-7H-benzo[de]anthracen-3-yl)amino)methyl)phosphonate* (**4c**) was obtained as a red solid in a 55% yield with a m.p. of 120 °C. Rf = 0.66 (hexane-acetone, *v*/*v* 1:2). ^1^H NMR (500 MHz, *Chloroform-d*) δ 8.79 (d, J = 7.3 Hz, 1H_Ar_), 8.38 (d, J = 7.9 Hz, 1H_Ar_), 8.35 (d, J = 8.3 Hz, 1H_Ar_), 8.10 (d, J = 8.2 Hz, 1H_Ar_), 8.06 (d, J = 8.2 Hz, 1H_Ar_), 7.81–7.70 (m, 2H_Ar_), 7.60 (t, J = 7.7 Hz, 1H_Ar_), 7.38 (t, J = 7.5 Hz, 1H_Ar_), 7.13–7.05 (m, 2H_Ar_), 6.36 (d, J = 8.2 Hz, 1H_Ar_), 5.81 (dd, J = 10.9, 6.3 Hz, 1H, NH), 4.97 (dd, J = 24.0, 5.9 Hz, 1H, NCH), 3.80 (d, J = 10.8 Hz, 3H, OCH_3_), 3.65 (d, J = 10.8 Hz, 3H, OCH_3_). ^13^C NMR (126 MHz, *Chloroform-d*) δ 183.94 (C=O), 161.89 (C), 159.30 (C), 143.01 (d, J = 13.7 Hz, C), 136.60 (C), 133.98 (dd, J = 8.4, 4.8 Hz, CH), 133.36 (CH), 132.44 (C), 132.43 (d, J = 5.2 Hz, CH), 130.26 (CH), 129.75 (C), 129.24 (C), 129.05 (CH), 128.83 (C), 128.24 (CH), 128.03 (CH), 126.72 (d, J = 55.9 Hz, CH), 125.80 (d, J = 16.5 Hz, CH), 123.42 (C), 122.12 (CH), 118.21 (C), 117.26 (dd, J = 20.0, 2.5 Hz, CH), 113.46 (C), 106.88 (CH), 54.56 (d, J = 151.5 Hz, NCH), 54.39 (d, J = 7.1 Hz, OCH_3_), 54.21 (d, J = 7.0 Hz, OCH_3_). ^31^P NMR (202 MHz, *Chloroform-d*) δ 22.74 (d, J = 5.2 Hz). FTIR (neat): 1648 (C=O); 2236 (C≡N); 2851, 2962, 3023 and 3067 (CH); 3297 (NH). ESI-FTMS: calculated for [C_27_H_20_FN_2_O_4_P+H+]: 487.1218, found: 487.1200.

### 3.3. Toxicology

For toxicological study, ethanol solutions of the obtained compounds **2a** and **4a**, at concentrations of 3 × 10^−3^ M, were prepared. Then, 10 mL of this solution was diluted with water to 500 mL to obtain a finely dispersed suspension with a resulting concentration of 6 × 10^−5^ M. Then, by means of dilution, a suspension with a concentration of 3 × 10^−6^ M was prepared. Within 24 h, the etiolated wheat sprouts of the cultivar “Fenomen” were grown in a dark chamber at a temperature of 26 °C on moist filter paper. A morphology test on filter paper was carried out in plastic dishes with three layers of paper on the bottom. Each dish contained 5 mL of dye solution or 5 mL of distilled water (control) and 30 seeds, covered by a lid. Dishes containing seeds were incubated for 5 days in a dark chamber at 26 ± 1 °C.

#### 3.3.1. Quantification of Malondialdehyde

The MDA content was determined by the thiobarbituric acid (TBA) reaction as described previously by Ali et al., with slight modifications [60]. Wheat germ, the first leaves, and the coleoptiles were homogenized in 0.1% trichloroacetic acid solution (1/10) and centrifuged for 15 min (14,000 rpm). After centrifugation, 2.5 mL of 0.5% thiobarbituric acid (in 20% trichloroacetic acid solution) was added to 1 mL of the upper fraction, and the solution was incubated in hot water (95 °C) for 30 min. The solution was cooled immediately to stop the reaction and centrifuged for 30 min (14,000 rpm). The optical density of the solution was determined at 532 nm and 600 nm wavelengths using a UV/VIS spectrophotometer, and the MDA concentration was calculated by subtracting the non-specific absorbance (600 nm) from the absorbance at a wavelength of 532 nm (ɛ = 155 mM^–1^cm^–1^).

#### 3.3.2. Electrolyte Leakage Measurements

The electrolyte leakage was determined as described previously by Guo et al. [61]. Organs of four seedlings (the first leaves, coleoptiles) were immersed in 15 mL deionized water (24 h, room temperature). After 24 h, the initial conductivity of the wheat germ organs was determined using a conductometer. The test tubes with the samples were placed into boiling water for 15 min, and after cooling to room temperature, the conductivity of the samples was again determined.

### 3.4. Imaging

Adult *Opisthorchis felineus* trematodes were collected within the framework of the project AP05131132, “PCR test for the detection and differential diagnosis of pathogens of opisthorchiasis and metorchiasis”. This research was approved by the Animal Ethics Committee of Veterinary Medicine Faculty of KATU (Ethical approval letter, No: 1, 9 November 2017).

An ethanol solution of the synthesized dyes with a molar concentration of 10^−4^ M was used for staining of the parasite sample for 10 min. Then, the dye was washed out three times with 70% ethanol. Further, the sample was placed into ethanol–xylene solution (1:1) for 10 min. Finally, the specimens were mounted on Canada balsam and covered with coverslips (24 × 24 mm).

A CLSM Eclipse Ti-E microscope, outfitted with a digital sight DS-U3 camera and configured with a high-speed multiphoton A1R MP confocal system and motorized stage (Nikon, Japan) was used. The CLSM images underwent processing using the NIS Elements Advanced Research 3.2 64-bit software (Nikon, Tokyo, Japan). Two lasers were employed to visualize the parasite: a 488 nm laser with a FITC filter (500–550 nm) and a 638 nm laser with a Cy5 filter (662–737 nm). The fluorescence signal registration was conducted using an internal spectral detector. The registration process began with a start wavelength set 20 nm higher than the excitation wavelength, extending to the edge of the red visible spectrum. No passive cutoff filters were inserted into the optical path. Images were captured as Z stacks with a 2.0 µm Z step size.

### 3.5. Single-Crystal X-ray Diffraction Analysis

The studied single crystals were grown from dichloromethane via slow evaporation. Diffraction data were collected at 160 K on a Rigaku, XtaLAB Synergy, Dualflex, HyPix diffractometer using monochromated Cu-Kα radiation (λ = 1.54184 Å).

For the compound **2a**, the crystal structure was solved using the heavy-atom method [62], and for the compound **4a**, the crystal structure was solved with the SIR2011 [63] structure solution program using Direct Methods Refined with the ShelXL [64] refinement package and least squares minimization. All nonhydrogen atoms were refined in anisotropic approximation. The hydrogen atoms involved in the formation of H-bonds were refined isotopically; all other H-atoms were refined using a riding model with Uiso(H) = 1.2Ueq(C).

Crystal Data for **2a** (C_23_H_19_BrNO_5_P; *M* = 500.27 g/mol): triclinic, space group *P*
$\overset{-}{1}$ (no. 2), *a* = 7.5626(2) Å, *b* = 7.7876(2) Å, *c* = 18.0152(5) Å, α = 85.582(2)°, β = 78.626(2)°, γ = 85.643(2)°, *V* = 1035.07(5) Å^3^, *Z* = 2, *T* = 150.0(3) K, μ(CuKα) = 3.759 mm^−1^, *Dcalc* = 1.605 g/cm^3^, 17,799 reflections measured (2Θ ≤ 160°), and 4456 unique (*R*_int_ = 0.0343, *R*_sigma_ = 0.0246) were used in all calculations. The final *R*_1_ was 0.0770 (*I* > 2σ(*I*)), and *wR*_2_ was 0.2016 (all data). For further details, see the crystallographic data for this compound deposited at the Cambridge Crystallographic Data Centre. Deposition number CCDC 2314862 contains the supplementary crystallographic data for this paper.

Crystal Data for **4a** (C_26_H_21_BrNO_4_P; *M* = 520.03 g/mol): monoclinic, space group *P*2_1_/*c* (no.14), *a* = 10.8242(3) Å, *b* = 8.2331(3) Å, *c* = 24.9042(5) Å, β = 93.006(2)°, *V* = 2216.3(1) Å^3^, *Z* = 4, *T* = 160.0(2) K, μ(CuKα) = 3.450 mm^−1^, *Dcalc* = 1.517 g/cm^3^, 22,314 reflections measured (2Θ ≤ 160°), and 4782 unique (*R*_int_ = 0.0738, *R*_sigma_ = 0.0629) were used in all calculations. The final *R*_1_ was 0.0541 (*I* > 2σ(*I*)), and *wR*_2_ was 0.1480 (all data). For further details, see the crystallographic data for this compound deposited at the Cambridge Crystallographic Data Centre. Deposition number (https://www.ccdc.cam.ac.uk/services/structures (accessed on 6 December 2023)) CCDC 2333330 contains the supplementary crystallographic data for this paper.

## 4. Conclusions

In conclusion, this study presents a comprehensive investigation into the synthesis, characterization, and potential applications of α-aminophosphonates derived from anthraquinone and benzanthrone. The synthesized compounds exhibited notable fluorescence properties with solvatochromic behavior, making them suitable for diverse applications. The spectroscopic analysis confirms the chemical structures, and the compounds’ luminescent properties are thoroughly explored in various solvents. The toxicity assessment on etiolated wheat sprouts reveals varying degrees of impact on growth and oxidative damage. Importantly, the research introduces these α-aminophosphonates as promising candidates for CLSM imaging of the parasitic flatworm *Opisthorchis felineus*, showcasing their potential in visualizing biological specimens. The X-ray crystallographic study of an anthraquinone α-aminophosphonate contributes valuable information regarding the compound’s molecular arrangement and intermolecular interactions. Overall, the synthesized compounds demonstrate potential for versatile applications, particularly in the investigation of parasitic organisms.

## Figures and Tables

**Figure 1 molecules-29-01143-f001:**
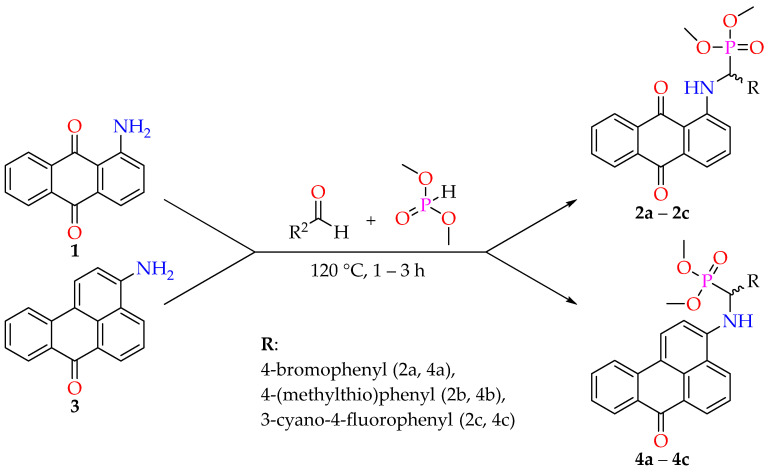
Synthesis of target α-aminophosphonates.

**Figure 2 molecules-29-01143-f002:**
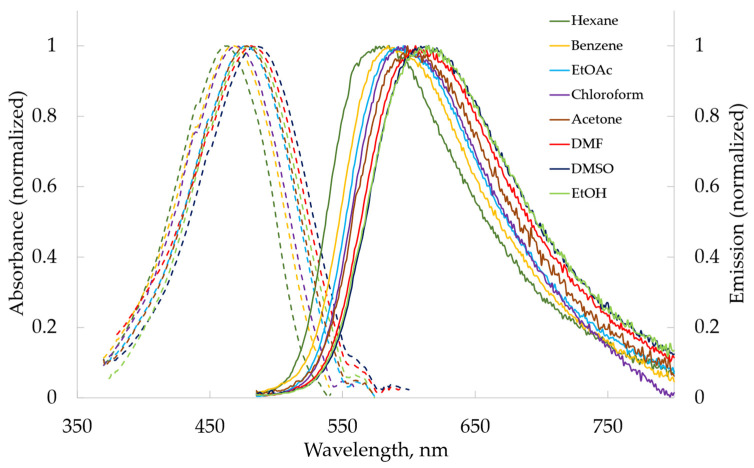
The UV–Vis absorption and fluorescence emission spectra of compound **2a** in various organic solvents.

**Figure 3 molecules-29-01143-f003:**
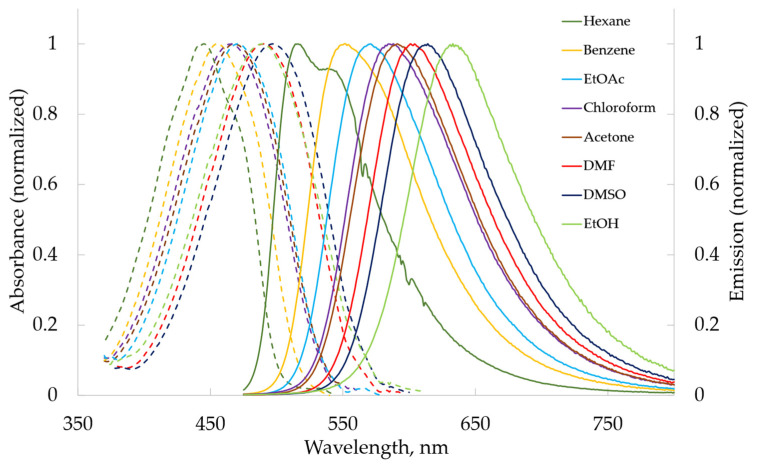
The UV–Vis absorption and fluorescence emission spectra of compound **4a** in various organic solvents.

**Figure 4 molecules-29-01143-f004:**
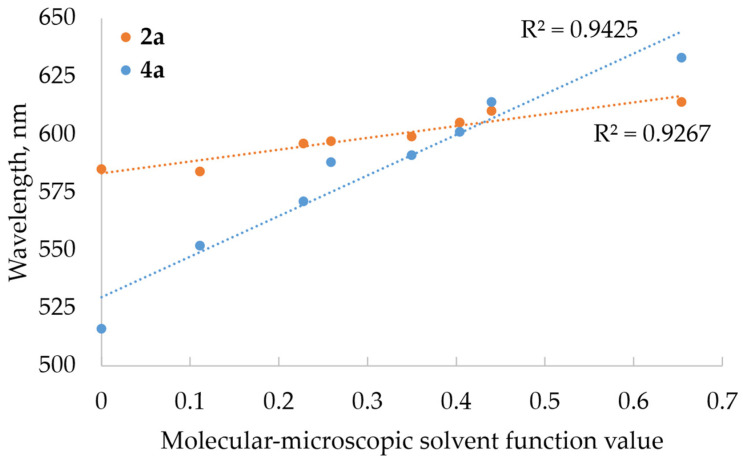
Emission wavelength maxima versus molecular–microscopic solvent polarity parameter ($E_{T}^{N}$) for compounds **2a** and **4a** in selected solvents; dotted lines represent linearity trend.

**Figure 5 molecules-29-01143-f005:**
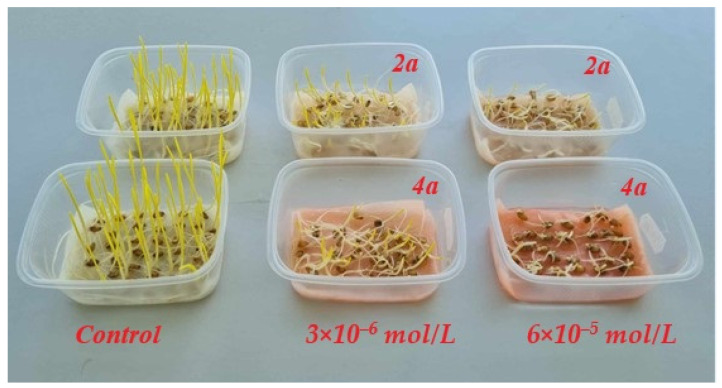
Wheat seedlings cultivated in the presence of dyes **2a** and **4a**, with concentrations 0 M, 6 × 10^−5^ M, 3 × 10^–6^ M.

**Figure 6 molecules-29-01143-f006:**
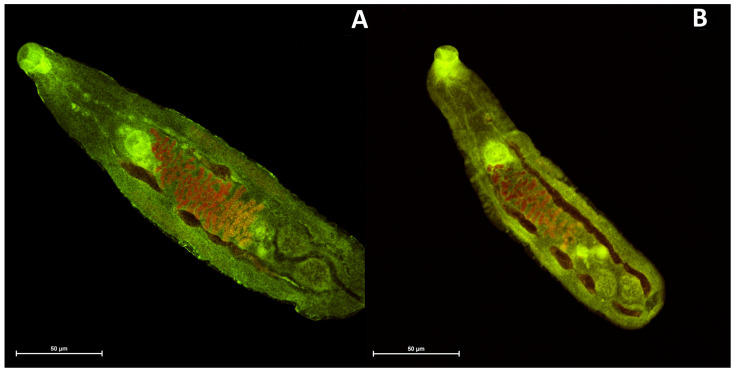
Adult *Opisthorchis felineus* stained with the examined dyes: (**A**)—**2b**; (**B**)—**2c**.

**Figure 7 molecules-29-01143-f007:**
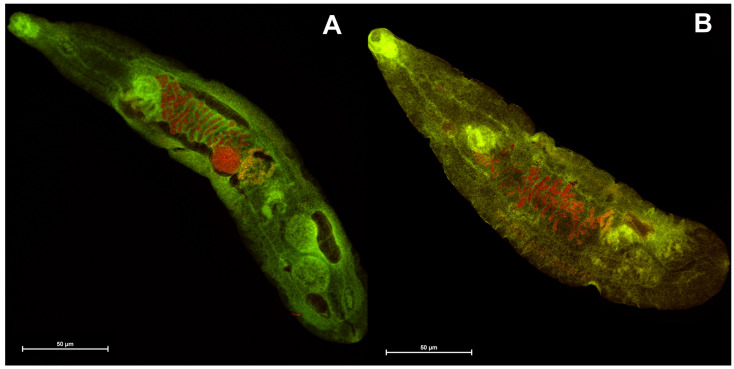
Adult *Opisthorchis felineus* stained with the examined dyes: (**A**)—**4b**; (**B**)—**4c**.

**Figure 8 molecules-29-01143-f008:**
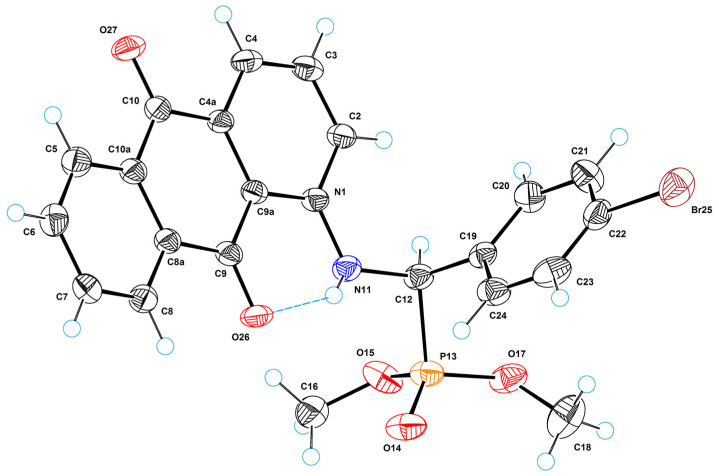
ORTEP diagram with the labels of atoms for **2a**.

**Figure 9 molecules-29-01143-f009:**
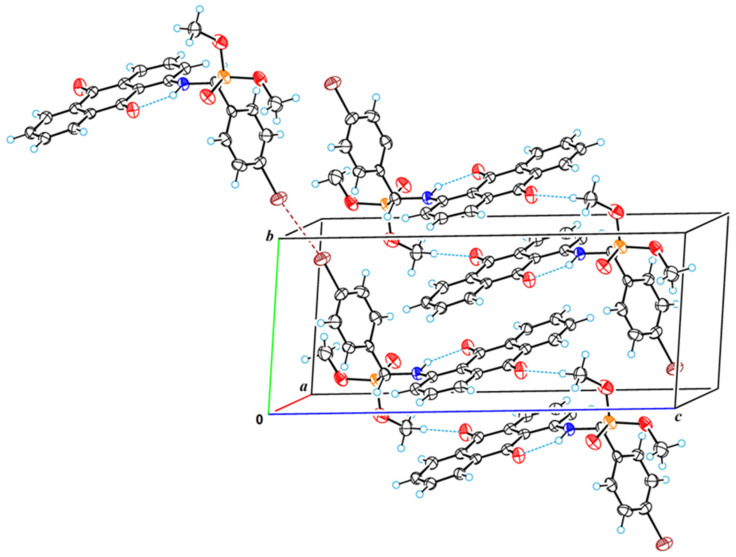
A fragment of the crystal structure of **2a**, showing hydrogen and halogen bonds. Red—oxygen, blue—nitrogen, orange—phosphorous, brown—bromine. Brown dashed line represents halogen bond between bromine atoms and blue dotted line represents hydrogen bonding.

**Figure 10 molecules-29-01143-f010:**
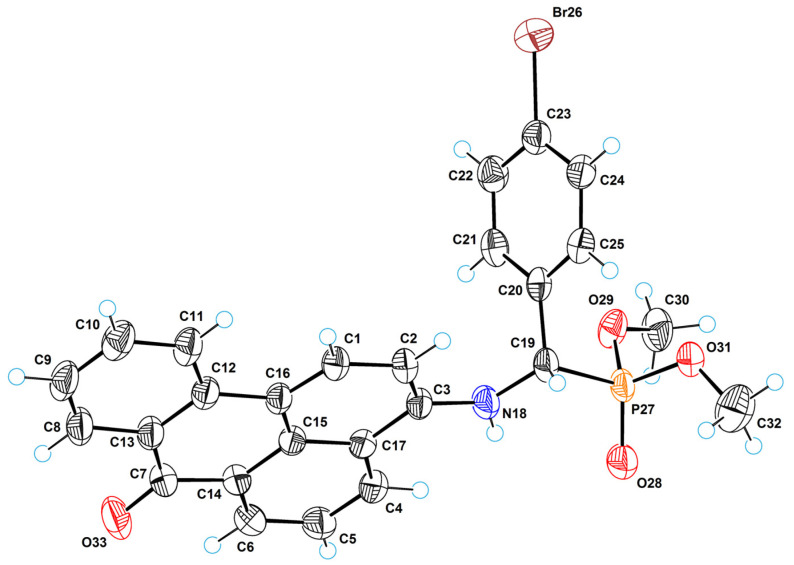
ORTEP diagrams with labels of atoms for **4a**. Red—oxygen, blue—nitrogen, orange—phosphorous, brown—bromine.

**Figure 11 molecules-29-01143-f011:**
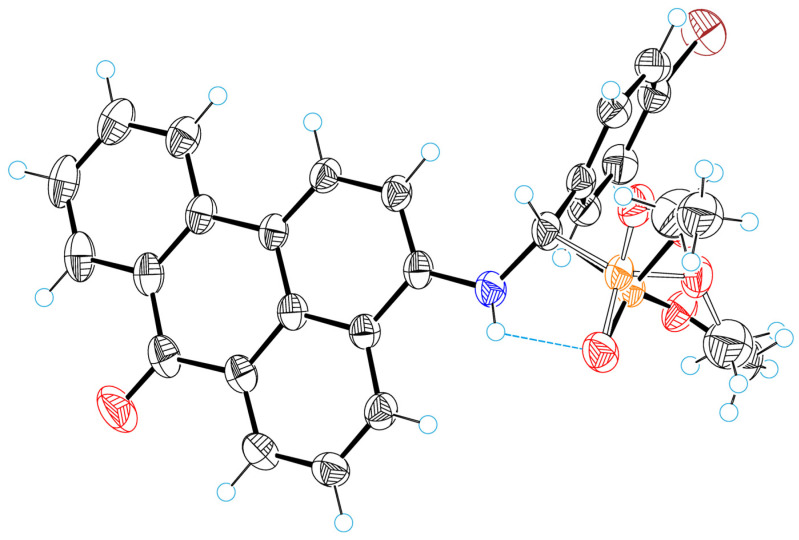
H-bond formation and disorder of the dimethyl phosphonate group in **4a**. Red—oxygen, blue—nitrogen, orange—phosphorous, brown—bromine.

**Figure 12 molecules-29-01143-f012:**
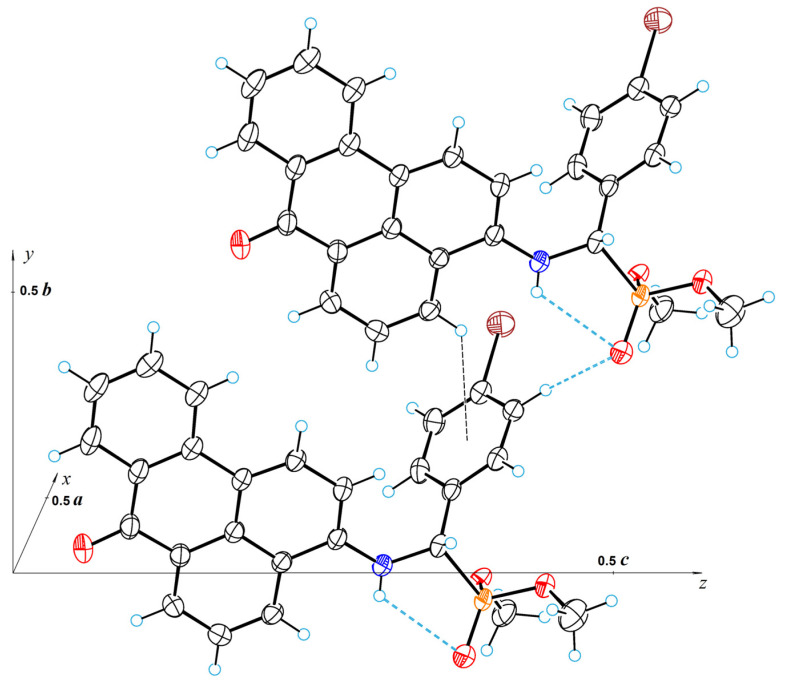
A fragment of the crystal structure of 4a, showing hydrogen and C4−H4···π bonds. Red—oxygen, blue—nitrogen, orange—phosphorous, brown—bromine.

**Table 1 molecules-29-01143-t001:** Absorption maxima (and extinction coefficients) of synthesized dyes in organic solvents (concentration: 10^–5^ M).

Solvent	λ_abs_ Max, nm (lgε)					
	2a	2b	2c	4a	4b	4c
Hexane	471 (3.99)	475 (3.84)	465 (3.60)	454 (4.35)	457 (4.13)	447 (3.76)
Benzene	478 (4.00)	481 (3.89)	471 (3.87)	465 (4.38)	467 (4.27)	459 (4.00)
Chloroform	480 (3.95)	483 (3.81)	448 (3.95)	476 (4.37)	467 (4.24)	465 (4.00)
EtOAc	477 (4.04)	480 (3.86)	472 (3.83)	470 (4.40)	472 (4.21)	465 (4.00)
Acetone	478 (4.02)	478 (3.85)	472 (3.84)	477 (4.42)	478 (4.26)	476 (4.00)
DMF	481 (4.04)	486 (3.86)	478 (3.83)	490 (4.37)	491 (4.27)	486 (4.04)
DMSO	484 (4.03)	488 (3.88)	479 (3.85)	497 (4.40)	498 (4.25)	491 (4.08)
EtOH	480 (3.92)	478 (3.93)	473 (3.83)	489 (4.23)	488 (4.23)	483 (4.04)

**Table 2 molecules-29-01143-t002:** Fluorescence maxima (and quantum yields) of synthesized dyes in organic solvents (concentration: 10^–5^ M).

Solvent	λ_em_ Max, nm					
	2a	2b	2c	4a	4b	4c
Hexane	585 (0.05)	-	-	516 (0.49)	555 (0.48)	562 (0.08)
Benzene	584 (0.11)	580 (0.17)	572 (0.03)	552 (0.47)	555 (0.57)	542 (0.18)
Chloroform	596 (0.07)	595 (0.09)	550 (0.14)	571 (0.41)	589 (0.48)	574 (0.22)
EtOAc	597 (0.08)	592 (0.07)	582 (0.02)	588 (0.48)	578 (0.53)	566 (0.18)
Acetone	599 (0.01)	604 (0.08)	593 (0.02)	591 (0.43)	597 (0.56)	587 (0.15)
DMF	605 (0.06)	604 (0.03)	606 (0.01)	601 (0.45)	611 (0.48)	610 (0.13)
DMSO	610 (0.04)	617 (0.03)	616 (<0.01)	614 (0.36)	620 (0.42)	619 (0.12)
EtOH	614 (0.02)	628 (0.05)	609 (<0.01)	633 (0.19)	636 (0.23)	629 (0.10)

**Table 3 molecules-29-01143-t003:** Stokes shift of synthesized dyes in organic solvents (concentration: 10^–5^ M).

Solvent	ν_abs_ − ν_em_, cm^–1^					
	2a	2b	2c	4a	4b	4c
Hexane	4137	-	-	2647	3864	4578
Benzene	3797	3549	3749	3389	3395	3336
Chloroform	4055	3897	4140	3495	4435	4084
EtOAc	4214	3941	4004	4270	3885	3838
Acetone	4226	4364	4323	4044	4170	3973
DMF	4261	4020	4419	3769	4000	4183
DMSO	4268	4284	4643	3834	3951	4212
EtOH	4547	4997	4721	4652	4769	4806

**Table 4 molecules-29-01143-t004:** Morphology of the wheat sprouts cultivated in the presence of dyes **2a** and **4a**.

	Length of First Leaf (mm)	Length of Coleoptile (mm)	Phytotoxicity on the First Leaf (%)	Phytotoxicity on the Coleoptile (%)
Control	68.26	34.52	-	-
**2a** (3 × 10^–6^ M)	20.52	12.70	70	63
**4a** (3 × 10^–6^ M)	24.24	14.76	64	57

**Table 5 molecules-29-01143-t005:** Electrolyte leakage and MDA assay of the wheat sprouts cultivated in the presence of dyes **2a** and **4a**.

	Electrolyte Leakage		MDA Assay	
	First Leaf	Coleoptile	First Leaf	Coleoptile
Control	41	15	31.9	15.1
**2a** (3 × 10^–6^ M)	29	23	63.8	40.9
**4a** (3 × 10^–6^ M)	20	21	67.0	46.5

## Data Availability

Data is contained within the article or Appendix A.

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
