# Peer review of "A Comparative Study of New Fluorescent Anthraquinone and Benzanthrone α-Aminophosphonates: Synthesis, Spectroscopy, Toxicology, X-ray Crystallography, and Microscopy of Opisthorchis felineus"

_molecules, 2024, doi:10.3390/molecules29051143_

Round 1
Reviewer 1 Report
Comments and Suggestions for Authors
This a lovely written and well-organized work on synthesis, spectral analysis, single crystal X-ray analysis of one compound and biological testing of anthraquinone and benzanthrone α-aminophosphonates. The X-ray structure is most in my field of expertise. Saying that and not detecting anything wrong, it would be helpful in a revision, if so to requested, to include a checkcif printout for the structure (complete with hkl-data and res file integrated).
Author Response
Thank you sincerely for your invaluable contribution and encouraging feedback on our publication, it is greatly appreciated.
We have also added information for crystal structure of compound 4a.
cif. files for both structures are added.
Reviewer 2 Report
Comments and Suggestions for Authors
1. It is unclear about what structures that author is talking about since the author have not labelled it correctly in Figure 1. The author need to labelled which a,b,c is referring to which R group.
2. Author should plot out all the UV–Vis absorption and fluorescence emission spectra (like figure 3) of 2abc and 4abc and placed in into the supporting information.
3. Since the compound is not labelled, the reviewer did not know which structures were the author talking about.
4. Why the author only choose 2a and 4a for study, please explain inside the text. the reviewer will suggest the author also perform the same experiment(toxicity) for 2bc and 4bc.
5. For the toxicity test, as the component will not dissolve in water, what solvent did the author used to dissolve the dye? (it is necessary to indicate in the manuscript) If so, the control experiment should also added the same organic solvent. If the compounds are water soluble, please include the NMR in D2O and study all the photophysical property in H2OO again.
6. Please also do the quantum yield measurement of all the synthesized dye.
7. The ESI in the supporting should include the simulated result and compared with the experimental pattern together.
8. Why only 2a give the crystal structure? please explain if it is not possible to grow in the text. please also indicate the condition and method for the crystal growing.
9. Xray structures should be uploaded to CCDC.
Comments on the Quality of English Language
no
Author Response
- It is unclear about what structures that author is talking about since the author have not labelled it correctly in Figure 1. The author need to labelled which a,b,c is referring to which R group.
We have now corrected the Figure 1.
- Author should plot out all the UV–Vis absorption and fluorescence emission spectra (like figure 3) of 2abc and 4abc and placed in into the supporting information.
Following guidance from another reviewer, we have included Figures 2 and 3, showcasing the spectra of compounds 2a and 4a. However, we have determined that adding spectra for compounds 2bc and 4bc would not contribute significantly to the study as information is already reported in Tables 1 - 3.
- Since the compound is not labelled, the reviewer did not know which structures were the author talking about.
We trust that the revised iteration of Figure 1 resolves any previous concerns on this matter.
- Why the author only choose 2a and 4a for study, please explain inside the text. the reviewer will suggest the author also perform the same experiment(toxicity) for 2bc and 4bc.
We acknowledge the suggestion to extend the experiment to include compounds 2b - c and 4b - c. However, it's important to note that our study was designed to investigate the general toxicological characteristics of anthraquinone and benzanthrone aminophosphonates and compare them, rather than exploring the influence of substituents on phenyl rings. While extending the experiment to include additional compounds could provide valuable insights into the broader scope of these chemicals, it would deviate from the primary objective of our study.
- For the toxicity test, as the component will not dissolve in water, what solvent did the author used to dissolve the dye? (it is necessary to indicate in the manuscript) If so, the control experiment should also added the same organic solvent. If the compounds are water soluble, please include the NMR in D2O and study all the photophysical property in H2OO again.
The manuscript now includes the previously omitted information addressing this ambiguity (Section 3.3.).
- Please also do the quantum yield measurement of all the synthesized dye.
Kindly refer to Table 2, as the information therein has already been furnished in the initial submission.
- The ESI in the supporting should include the simulated result and compared with the experimental pattern together.
Regrettably, conducting simulations falls outside the scope of our expertise and the focus of this research.
- Why only 2a give the crystal structure? please explain if it is not possible to grow in the text. please also indicate the condition and method for the crystal growing.
We have now added crystal structure information for 4a as well. Method for crystal growing is now added.
- Xray structures should be uploaded to CCDC.
CCDC numbers are now provided for both compounds.
Reviewer 3 Report
Comments and Suggestions for Authors
The authors compared the photophysical properties of fluorescent anthraquinone α-amino phosphonates and 2-benzanthrone α-amino phosphonates derivatives and applied them to confocal laser scanning microscopy (CLSM) of a parasite. The authors also studied their solvatochromism ability. The manuscript is interesting; however, the structure elucidation spectra are missing, and some aspects need to be addressed.
1- Please avoid using abbreviations such as CLSM in the title.
2- The authors should supply FT-IR, NMR, and MS spectra as supporting information.
3- In Figure 1, avoid using R1 and R2 as R1 is always methyl. You can use only one R, which is that of aldehyde.
4- The authors should clearly state the novelty of the manuscript over their previously published papers related to these compounds (Ref 1 and 2).
5- The UV–Vis absorption and fluorescence emission spectra of compound 4a in various organic solvents should be added to the manuscript and compared in detail with Figure 2.
6- The authors should explain the reason for the elevated toxicity of 2a vs. 4a. The reason for this phenomenon could be the quinone nature of 2a, which, upon reaction with antioxidants, releases ROS. The authors can refer to this reference (10.1021/acs.analchem.0c00044).
7- Please summarize the discussion about the imaged trematode (lines 217 – 240).
8- The authors should supply an image for the trematode using classical KI dyes.
Author Response
1- Please avoid using abbreviations such as CLSM in the title.
Corrected to ‘microscopy’.
2- The authors should supply FT-IR, NMR, and MS spectra as supporting information.
Kindly refer to Supporting information file, as the information therein has already been furnished in the initial submission.
3- In Figure 1, avoid using R1 and R2 as R1 is always methyl. You can use only one R, which is that of aldehyde.
Corrected.
4- The authors should clearly state the novelty of the manuscript over their previously published papers related to these compounds (Ref 1 and 2).
We have revised the final paragraph of the Introduction section to enhance clarity and emphasize the novelty of our message.
5- The UV–Vis absorption and fluorescence emission spectra of compound 4a in various organic solvents should be added to the manuscript and compared in detail with Figure 2.
Manuscript now contains spectra for both compounds 2a and 4a.
6- The authors should explain the reason for the elevated toxicity of 2a vs. 4a. The reason for this phenomenon could be the quinone nature of 2a, which, upon reaction with antioxidants, releases ROS. The authors can refer to this reference (10.1021/acs.analchem.0c00044).
Explanation is now added.
7- Please summarize the discussion about the imaged trematode (lines 217 – 240).
Discussion is now summarized.
8- The authors should supply an image for the trematode using classical KI dyes.
Apologies for any confusion caused; the abbreviation "KI" originally referred to benzanthrone dyes as a code, and regrettably, we overlooked updating the code for the compound number, which has now been corrected.
Round 2
Reviewer 2 Report
Comments and Suggestions for Authors
The author did the revision accordingly, I would like to suggest it to be accepted after some small minors point.
Still, the author need to simulate the ESI-MS isotopic pattern of the ESI-MS spectrum and do the comparsion, it could be done by the MS software.
Comments on the Quality of English Languageno
Author Response
Unfortunately, we couldn't conduct the analysis due to limitations. Mass spectra data was received from another institute, and we do not have the necessary software for simulations. That team is busy with other projects, making it impractical to allocate resources to this task. Also, outsourcing simulations introduces uncertainties in timelines and deliverables, potentially extending the process indefinitely. We appreciate your understanding regarding this matter. Thank you for your continued support and consideration.
Reviewer 3 Report
Comments and Suggestions for Authors
The authors have addressed the reviewer's comments adequately and the manuscript could be accepted in its current form.
Author Response
Thank you sincerely for your invaluable contribution and feedback on our publication, it is greatly appreciated.